# *XBP1* expression in pancreatic islet cells is associated with poor glycaemic control especially in young non-obese onset diabetes across ancestries

## Abstract

**Background** Individuals of South and East Asian ancestry have a higher risk of type 2 diabetes, often driven by insulin deficiency due to impaired beta-cell function. The transcription factor *XBP1* supports beta-cell survival by reducing cellular stress, but its role in diabetes risk and glucose regulation remains unclear. This study aimed to evaluate the impact of *XBP1* expression on diabetes risk, beta-cell function, glycaemic traits, and treatment response across ancestries.

**Methods** We performed colocalisation analyses to test whether *XBP1* expression in pancreatic islets and type 2 diabetes share causal variants. A lead variant regulating *XBP1* expression was identified and analysed in two South Asian cohorts from India to assess associations with beta-cell function and glucose levels. We further assessed glycaemic control using HbA1c in cohorts of British South Asians and white Europeans. We examined the effect of the variant on drugs designed to improve insulin secretion.

**Results** *XBP1* expression colocalises with diabetes risk in East Asians but not in white Europeans, and lower expression is associated with higher risk of diabetes. The lead SNP of the eQTL (rs7287124) is more common in East (65%) and South Asians (50%) compared to white Europeans (25%). rs7287124 is associated with lower beta-cell function using HOMA-B ($P = 5 \times 10^{-3}$, $n = 470$). In trans-ancestry meta-analyses rs7287124 is associated with 4.32 mmol/mol (95% CI: 2.60–6.04, $P = 8 \times 10^{-7}$) higher HbA1c. In individuals with young, non-obese onset diabetes, the trans-ancestry effect is 6.41 mmol/mol ($P = 2 \times 10^{-4}$). Variant carriers show impaired response to sulphonylureas.

**Conclusions** *XBP1* expression is associated with diabetes risk with particular value in under-represented populations at risk of young, non-obese onset diabetes.

## Plain language summary

Type 2 diabetes is a condition in which the body can have too much sugar in the blood. One pathway for Type 2 diabetes development is when beta-cells in the pancreas can no longer always function appropriately. This study focused on a gene called *XBP1*, which helps beta-cells function appropriately. We found that people with a common change in *XBP1* had poorer beta-cell function, worse blood sugar control when diagnosed with diabetes, and reduced response to sulphonylurea medications that are used to treat diabetes. These changes in *XBP1* are observed more often in individuals of South and East Asian or African ancestry, and the effect of the change is particularly strong in those diagnosed young and at a healthy weight. These findings highlight the need for ancestry-informed research and suggest that boosting *XBP1* expression in pancreatic cells could be a future strategy to improve diabetes care and outcomes.

The pathophysiology of type 2 diabetes mellitus (T2DM) is multifactorial[1]. The presentation of clinical features at diagnosis varies from those related to insulin resistance to insulin deficiency and mild age-related diabetes[1,2]. While white Europeans show higher proportion of insulin resistance-related diabetes, individuals of South Asian, East Asian, and African ancestry have a higher proportion of insulin deficiency[3–6]. This, in part, leads to the higher frequency of young-onset diabetes in people of Asian and African ancestry[7]. Insulin deficiency is driven by pancreatic beta-cell dysfunction, which could

be due an insufficient number of beta-cells, reduced mass, function, early cell death, increasing visceral fat, or perhaps a combination of these factors.

The efficiency of pancreatic beta-cells depends on a highly competent endoplasmic reticulum (ER) due to the high amount of protein synthesis and secretion required, especially in high-demand states[8]. The unfolded protein response (UPR) is a key cellular process reversing ER stress[9,10]. Therefore, an unimpaired UPR is crucial for continued insulin synthesis throughout life. Persistent ER stress, caused by impairments in UPR, has

✉e-mail: moneeza.siddiqui@qmul.ac.uk

been linked to beta-cell dysfunction and death, while a functional UPR promotes increased insulin secretion[9,11–13].

XBP1 is a highly conserved transcription factor produced in response to ER stress, and its function is to regulate the UPR[14,15]. Impairment of the UPR is linked to a decline in pancreatic islet function. Islets of eight individuals with T2DM showed decreased spliced *XBP1* expression[16]. We elucidate this mechanism in **the visual abstract**. In 2022, Lee et al. showed that knocking out *XBP1* in high-fat diet-fed mice resulted in the development of T2DM through failed beta-cell compensation and increased apoptosis[17]. Pancreatic beta-cells in *XBP1* knock-out mice showed greater beta-cell dedifferentiation, beta-to-alpha cell transdifferentiation, and impaired proinsulin processing. Together this rendered the beta-cells ineffective when challenged with metabolic stressors, leading to reduced beta-cell function and eventually hyperglycaemia. In 2023, Wang et al. used human single-cell sequencing and multi-omic experiments to identify mechanisms of beta-cell dysfunction in type 2 diabetes. They identified XBP1 as a key transcription factor that was down-regulated in individuals with diabetes compared to those with pre-diabetes and healthy controls, which sensitizes beta-cells to ER stress.

The genetics of beta-cell function, specifically insulin secretion, in humans is not as well studied as other aspects of T2DM. This is largely because most genetic studies of diabetes have been performed in well-powered cohorts of European ancestry, who have a lower burden of beta-cell dysfunction compared to other ancestries[18]. Furthermore, the largest meta-genome-wide association studies (GWAS) on beta-cell function were performed in healthy white Europeans[19] and remain the only contributor even in recent studies[20]. To address this gap in both data and knowledge, it is imperative to build on biological evidence of beta-cell susceptibility pathways and test their role in T2DM in non-white populations using approaches other than GWAS. Summary statistics from the most recent trans-ancestry genome-wide association study of T2DM reported a credible set of 126 variants in the genomic region on chromosome 22, but so far, none have been implicated with *XBP1* expression[21]. The role of *XBP1* expression in maintaining glucose homeostasis, glycaemic control, and response to diabetes therapeutics across human populations has not been previously studied.

We test the hypothesis that a SNPs associated with the expression of *XBP1* in pancreatic islet cells drives an increased risk of T2DM, reduced beta-cell function (HOMA-B), impaired pre-treatment glycaemic control (HbA1c), and impaired response to insulin secretagogues. We find that decreased *XBP1* expression in pancreatic islet cells is associated with increased type 2 diabetes risk, and that the variants associated with reduced *XBP1* expression are more common in East and South Asians as well as African populations. Using data from South Asians cohorts, we demonstrate that the variant is associated with lower beta-cell function and higher stimulated response. In trans-ancestry meta-analyses, we observe the effect of this variant on higher HbA1c. Finally, we demonstrate that drugs like sulphonylureas are less likely to work in carriers of these variants.

## Methods
To aid readers, we have included table with acronyms for study names and details in Supplementary Table 1. Throughout the manuscript, where individual-level data are presented, we have used genetically-inferred ancestry.

### Identifying potentially lead variant for pancreatic islet expression of *XBP1*
To identify a Single Nucleotide Polymorphism (SNP) affecting the expression of *XBP1* (rs7287124, eSNP), we used cis-eQTLs results from pancreatic islets ($n = 420$) and pancreatic beta-cell samples ($n = 26$) from the Integrated Network for Systematic analysis of Pancreatic Islet RNA Expression (InsPIRE) study (details in Supplementary Methods 1). Previous studies have estimated that the most significant variant in cis-eQTL analyses has a causal role 50% of the time (including when the causal variant has not been genotyped or imputed)[22]. This cis-eQTL was also replicated by the

Translational Human Pancreatic Islet Genotype tissue-Expression Resource (TIGER)[23]. A study flowchart is provided in Supplementary Fig. 1.

### Assessing the probability of the lead variant being associated with type 2 diabetes using colocalisation analyses
We performed colocalisation analyses to test if the lead variant identified (rs7287124) was associated with both expression of *XBP1* and type 2 diabetes risk. Colocalisation analyses compare the distribution of summary statistics from large-scale association studies of complex traits and eQTLs, while reducing false positive discoveries by correcting for linkage disequilibrium across a genetic region[24,25]. For colocalisation analyses, we used the programme, ezQTL, which computes the probability of colocalisation using two established methods, HyPrColoc (Hypothesis Prioritisation in multi-trait Colocalization)[26] and eCAVIAR (CAusal Variants Identification in Associated Regions)[27]. Since colocalisation analyses were used to check for linkage contamination, only single-ancestry GWA studies could be utilised. Therefore, we used summary statistics from the two largest available ancestry-specific GWAS of T2DM: one of white Europeans ($n$ cases = 80,154) and the another of East Asian descent ($n = 36,614$) from the DIAMANTE consortium and Biobank Japan, respectively[21,28]. Unfortunately, a similarly powered GWAS in south Asians is not available, and therefore colocalisation with south Asians was not undertaken. We then assessed the direction of effect between expression and T2DM risk using inverse variance weighted and weighted median methods in the MR-Egger package[29].

All resources used for summary statistics have been published previously and are publicly available, additional details on these are provided in Supplementary Table 1 and Methods 1.

### Association of *XBP1* expression with glycaemic traits
We used individual-level data from 4 biobanks in this analysis: (1) the Tayside Diabetes Cohort based in Scotland, UK (TDC, comprised of white Europeans), (2) Dr. Mohan's Diabetes Specialities Centre (DMDSC) and (3) Telemedicine Project for Screening diabetes and complications in rural Tamil Nadu (TREND), both comprised of South Asian Indians based in southern India, and finally (4) Genes & Health (G&H, British South Asian Pakistani and Bangladeshi) based in London, UK. Study design and data collection for these biobanks have been described previously, and additional details in Supplementary Methods[30–32]. Briefly, TDC and G&H are cohorts with record linkage of electronic health data from their local National Health Service boards, which have been used for ancestry-specific diabetes research[3,33]. DMDSC has record-linked clinical data from visits to a private diabetes clinic and hospital chain in South India. In cohorts with linked health records: TDC, G&H, and DMDSC, only participants with newly diagnosed T2D (glycaemic measures 12 months before and 1 month after diagnosis) were included. In TREND, a survey of diabetes prevalence in rural South India, only participants with screen-detected diabetes were considered. Additional details about the cohorts, including genotyping methods, are provided in Supplementary Methods. All studies used Standard criteria for diagnosis of T2DM using fasting or random plasma glucose, Oral Glucose Tolerance Test (OGTT), or glycated haemoglobin (HbA1c). Serum C peptide (where available), fasted and stimulated glucose, and HbA1c were restricted to measures taken 12 months prior to, and up to, 1 month after diagnosis of T2DM to reduce contamination of effects by diabetes therapies.

### Glycaemic traits
Serum C-peptide and plasma glucose were measured only in the DMDSC bioresource and using a fixed protocol, fasted and stimulated measures[34]. Patients visit the facility in a fasted state, and a blood draw is made. These are then used in Homeostatic Models of Assessment (HOMA) to estimate pancreatic beta-cell function and insulin sensitivity[35,36]. Similarly, we assessed the effect of the eQTL variant on stimulated blood glucose (SBG, 2hrG) levels in rural-dwelling Asian Indians who were screen-detected as having diabetes during the TREND survey.

HbA1c is commonly recorded and provides a measure of long-term blood sugar control. We tested the association of the variant rs7287124 with HbA1c from all four biobanks. For meta-analyses of the effect of rs7187124 with HbA1c in people without diabetes, we utilised summary estimates from the Meta-Analysis of Glucose and Insulin-related traits Consortium (MAGIC). Summary estimates from the three largest ancestry-specific contributors: Europeans(70%), East Asians(13%), and South Asians were used[20].

## Sub-group analyses

Lower *XBP1* expression impacts beta-cell viability under high metabolic stress[17,37]. We hypothesised that individuals with lower expression of *XBP1* would be susceptible to metabolic stress, leading to a younger age of T2DM onset with non-obese BMI. Therefore, we expect that the effects of the lead SNP would vary by BMI and age at diagnosis, and this would be detectable in an interaction term between the two variables when predicting the association between the lead SNP and glycaemic traits in the full population.

Thresholds for non-obese and obese BMI followed ethnicity-specific guidelines; non-obese for white Europeans was BMI <30 kg/m$^2$, whereas for South Asian Indians <25 kg/m$^2$, and South Asian Pakistanis and Bangladeshis living in the UK, <27.5 kg/m$^2$[38,39]. Associations between the *XBP1* lead SNP (rs7287124) were tested in the full cohort, and in those with (1) young onset and non-obese BMI at diagnosis, and (2) in those with older onset and obese BMI. Genetic data from these cohorts have been used previously, and an overview of genotyping methods is provided in Supplementary Methods 3.

## Statistical models used

For HOMA-B associations, we used the following 3 models: one unadjusted model assessing genotype effect on beta-cell function, second adjusted for HOMA-S (insulin sensitivity), age, BMI, and sex, and a third to determine if there was an interaction between the variant, age, and BMI at diagnosis. For stimulated blood glucose, models were adjusted for BMI and age at diagnosis, as well as sex. Finally, for HbA1c association testing, the full model was:

HbA1c ~ rs7287124 + Age at diagnosis + BMI at diagnosis + Sex + Age at diagnosis* BMI at diagnosis

Sub-group analyses models were not adjusted for the interaction, the model below was used:

HbA1c ~ rs7287124 + Age at diagnosis + BMI at diagnosis + Sex

All analyses were undertaken using R4.2.1[40]. We meta-analysed the overall and sub-group stratified associations assuming random-effects using the R package Metafor[41].

## Pharmacogenomic effects on sulphonylurea and GLP1-RA response

Finally, to test the hypothesis that *XBP1* expression exerts an effect on glycaemic control through beta-cell function, we used the largest available pharmacogenomic studies from the MetGen Plus and the DIRECT consortia to examine the effect of rs7287124 on response to an incretin mimetic like glucagon-like peptide 1- receptor agonist (GLP-1RA) and insulin secretagogues like sulphonylureas[42,43].

**Tayside, Scotland UK**. Ethical approval for the study was provided by the Tayside Medical Ethics Committee (REF:053/04) and the study has

been carried out in accordance with the Declaration of Helsinki. All volunteers provide written informed consent, allowing analysis of data for biomedical research and publication of results. Data access was through application.

**Genes and Health, UK**. Ethical approval is from the National Research Ethics Committee (London and Southeast), and the Health Research Authority (ref. 13/LO/124), and Queen Mary University of London is Sponsor. All volunteers provide written informed consent, allowing analysis of biomedical data and publication of results. Data access was through application.

All other studies and cohorts used (as summarised in Supplementary Table 1) are publicly available and links to publications have been provided.

## Ethical committee approvals

**DMDSC and TREND**. NIHR Global Health Research Unit on Global Diabetes Outcomes Research, Institutional Ethics Committee of Madras Diabetes Research Foundation, Chennai, India. IRB number IRB00002640. Granted 24th August 2017. All volunteers provide written informed consent, allowing analysis of data for biomedical research and publication of results. Data access was through application and as part of the NIHR Global Health Research Unit.

## Results

### rs7287124 is the lead eQTL variant for XBP1 expression in pancreatic islets

A single intronic variant; rs7287124 (Chromosome 22: 29239157A > G) was reported to act as a *cis*-eQTL for *XBP1* expression in pancreatic islets ($\beta_{\text{expression}} = -0.18$, $p = 2 \times 10^{-4}$, Table 1) by the InsPIRE study[44]. The direction of expression effect was consistent in pancreatic beta-cells eQTL results ($N = 26$, $\beta_{\text{expression}} = -0.219$, $p = 0.54$). Using TIGER summary statistics, we confirm that the G allele is associated with lower *XBP1* expression (Table 1 and Supplementary Table 2) and is not an eQTL for any other genes in the pancreas. Further, we observed that the effect of rs7287124 in both Biobank Japan[28] and the DIAMANTE trans-ancestry meta-GWAS summary estimates[21], is directionally consistent with increased diabetes risk (Table 1). The effect allele frequency (EAF) in European populations is 24%, whereas South Asians and East Asians have higher EAF of 54% and 65% respectively, suggesting that these ethnicities will have lower genetically predicted *XBP1* levels in pancreatic islets.

### rs7287124 is associated with *XBP1* expression and type 2 diabetes risk in East Asians but not white Europeans

We found a strong posterior probability that the same locus is associated with T2DM risk and *XBP1* expression in East Asians (islets = 1%, pancreas 5%) (Fig. 1a), while weaker probabilities were observed in white Europeans (islets = 0.3%, pancreas = 0.55%) using the eCAVIAR method. The HyPrColoc method showed probabilities of 0.88 and 0.87 in islets and pancreas for East Asians, and 0 for both tissues in Europeans. (Supplementary Figs. 2–4 and Table 3). In East Asians, the variant identified in the colocalisation analyses was rs58004020; however, this variant is not present in build 37 of the human genome for South Asians. Therefore, we identified a proxy variant that is in strong LD with this lead variant and is uniformly available across all ancestry

**Table 1 | Allele frequencies across ancestries and effects of lead eQTL variant on XBP1 expression in pancreatic beta-cells, T2DM risk**

| Variant | Chr: base position | Beta Islet Expression (InsPIRE), *P* value | Z score, *P* value Islet Expression (TIGER) | 1000 G European EAF (G) | 1000 G South Asian EAF (G) | 1000 G African EAF (G) | 1000 G East Asians EAF(G) | Beta T2DM (Biobank Japan) | Beta Diabetes (DIAM ANTE) |
|---|---|---|---|---|---|---|---|---|---|
| rs7287124 A > G | 22: 29239157 | −0.18, 2.9 × 10$^{-4}$ | −4.53, 5.8 × 10$^{-6}$ | 0.24 | 0.54 | 0.45 | 0.65 | 0.04, $p = 4.6 \times 10^{-6}$ | 0.02, $p = 1.4 \times 10^{-4}$ |

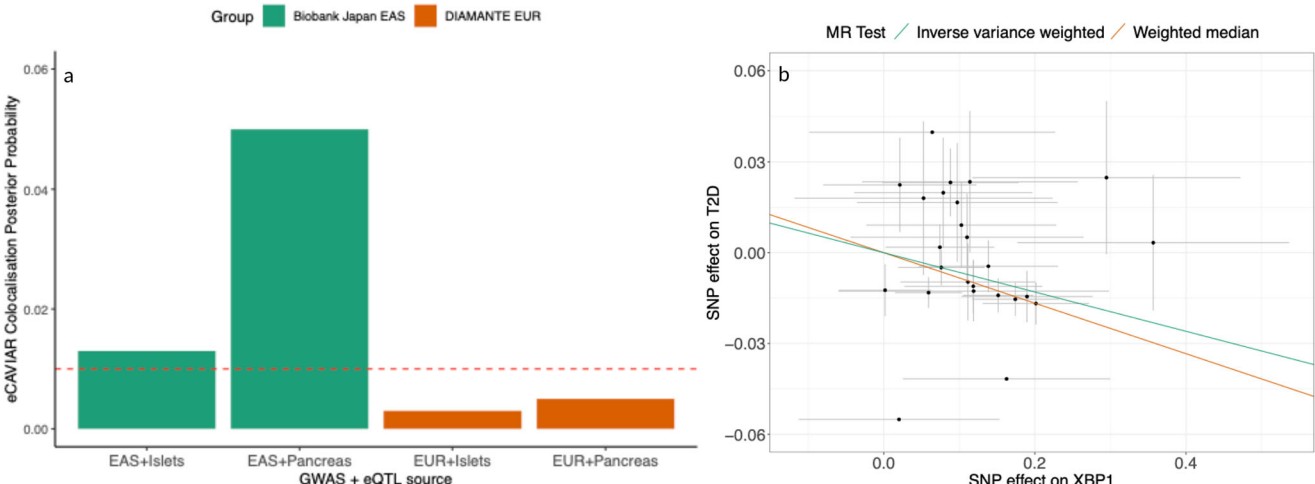

**Fig. 1 | Colocalisation and Mendelian Randomisation methods for XBP1 expression and type 2 diabetes risk. a** Colocalisation probabilities using eCAVIAR and HyPrColoc between *XBP1* expression in both pancreatic islets and GTEx pancreas samples and type 2 diabetes (T2D) risk in East Asians (EAS) from Biobank Japan and white Europeans (EUR) from DIAMANTE. Orange line indicates eCA-VIAR recommended threshold for significant colocalisation. In both cases, East Asians are more likely to have overlapping causal variants. **b** Mendelian Rando-misation (MR) ratio estimates of association between *XBP1* expression in pancreatic islets and T2D risk in Biobank Japan. These plots have been made using 25 variants with LD < 0.4 in the XBP1 region $+/-5KB$. Weighted median estimate for the effect was $-0.083$ (SE: 0.025), $P$ value $= 9.2 \times 10^{-3}$, and inverse variance weighted estimate was $-0.065$ (SE: 0.020), $P$ value $= 7.17 \times 10^{-4}$.

groups and has survived LD-pruning in GWAS summary statistics. The eQTL variant rs7287124 is in LD with rs58004020 in East Asians, Europeans, and South Asians: $R^2 = 0.99, 0.98$, and 0.98 respectively and $D' = 1, 0.72$, and 0.85, respectively. Both variants are in the gene *ZNRF3* but have no effect on the expression of *ZNRF3* in the pancreas or pancreatic islets. Since rs7287124 is the lead SNP of the eQTL and is in complete LD with rs58004020 in East Asians, where it was identified, rs7287124 was selected for downstream investigation with glycaemic traits and drug response. Therefore, colocalisation analyses strongly suggest that *XBP1* expression mediates the effect of rs7287124 on T2DM risk.

We then investigate the direction of effect by testing the hypothesis that lower *XBP1* expression in pancreatic islets was associated with higher T2DM risk. Using inverse variance weighted and weighted median models, we find this association between the two traits to be significant (B $-0.065$, SE 0.02, and $P < 7.17 \times 10^{-4}$) and directionally consistent with our hypothesis (Fig. 1b).

**Association with human beta-cell function in Asian Indians**
Baseline characteristics of cohorts with individual-level data are described in Supplementary Table 4. In 470 participants of the Asian Indian cohort, rs7287124 was associated with HOMA-B in unadjusted (Fig. 2a) and adjusted models (Supplementary Table 5). In additive models of genetic effect adjusted for HOMA-S (insulin sensitivity), age, BMI, and sex, the G allele was associated with lower HOMA-B (B: $-0.14$, SE: 0.05, $P = 5 \times 10^{-3}$). An interaction between age and BMI at diagnosis on HOMA-B was observed ($P = 4.3 \times 10^{-5}$). In individuals diagnosed young and non-obese, this effect was greater ($n = 82$, B: $-0.30$, SE: 0.14, $P = 0.036$). However, the effect was not significant in the older and non-lean sub-group ($n = 134$, $P = 0.27$). Recessive models were run to confirm the genetic model is additive (Supplementary Table 5). The variant was not associated with insulin resistance or sensitivity ($P = 0.12$).

**Association with stimulated blood glucose levels**
We then assess the association between rs7287124 and stimulated 2-hour blood glucose (SBG) levels in 484 rural-dwelling Asian Indians from TREND. The variant was associated with higher SBG in models adjusted for BMI, age, and sex. ($B = 0.7$ mmol/L, SE = 0.3, $P = 0.02$) (Fig. 2b) However, no effect was observable in either of the sub-groups.

**Association with HbA1c levels, especially in individuals diagnosed young and with non-obese BMI**
We then tested the hypothesis that the variant will likely be associated with longer-term glycaemic control using HbA1c, which is much more routinely measured. To assess the effect of rs7287124 across ancestries, we included individual-level data from newly-diagnosed South Asian Indians represented by DMDSC ($n = 459$) and TREND ($n = 471$), South Asian Bangladeshi and Pakistani represented by Genes and Health ($n = 644$), and white Europeans from the TDC (4908). Additionally, we used summary statistics from existing GWAS in individuals without T2DM.

The variant was in Hardy-Weinberg equilibrium in all cohorts. The trans-ancestry meta-analyses showed a significant increase in HbA1c per risk allele regardless of diabetes status (Fig. 3a). Random-effects meta-analysis showed the pooled effect across T2DM status to be 4.32 mmol/mol per risk allele (95%CI: 2.60, 6.04), $P$ value $8 \times 10^{-7}$. Therefore, homozygous carriers of the G allele are estimated to have 8.64 mmol/mol higher HbA1c levels compared to non-carriers. In meta-analyses of the effect in healthy individuals, each risk allele was associated with 5.37 mmol/mol higher HbA1c (95%CI:0.22, 10.52), while in people with T2DM, the effect was 4.37 (95% CI: 2.60– 6.55).

Approximately 7% of individuals ($n = 477$) with newly-diagnosed diabetes had young onset (<40 years) with non-obese BMI. Despite using ethnicity-specific BMI thresholds, this proportion differed across cohorts: in Asian Indians, 30% in DMDSC, 34% in TREND (<25 kg/m$^2$), in British South Asians from G&H, 9.5% (<27.5 kg/m$^2$), and in white Europeans from TDS, 2.4% (<30 kg/m$^2$). The meta-analysis in sub-groups shows that the effect of this variant is driven by the young, non-obese sub-group, as the per allele effect was 6.41 mmol/mol (3.04, 9.79), $P = 2 \times 10^{-4}$ (Fig. 3b). In contrast, those diagnosed older with obese BMI ($n = 3382$) showed no genetic effect ($P = 0.97$). In our study cohorts, 25% of South Asian Indians (DMDSC + TREND), 20% of South Asian Pakistani and Bangladeshi (G&H), and 0.05% of white Europeans (TDC) were homozygous carriers of the risk allele (G/G) (Supplementary Fig. 5).

**rs7287124 is associated with worse response to sulphonylureas but no differential response to GLP1-RA**
We hypothesized that carriers of an allele associated with reduced beta-cell function are likely to have limited response to insulin secretagogues. To test this hypothesis, we examined the effect of these variants on glycaemic response to sulphonylureas and GLP-1RA. Carriers of the G allele showed

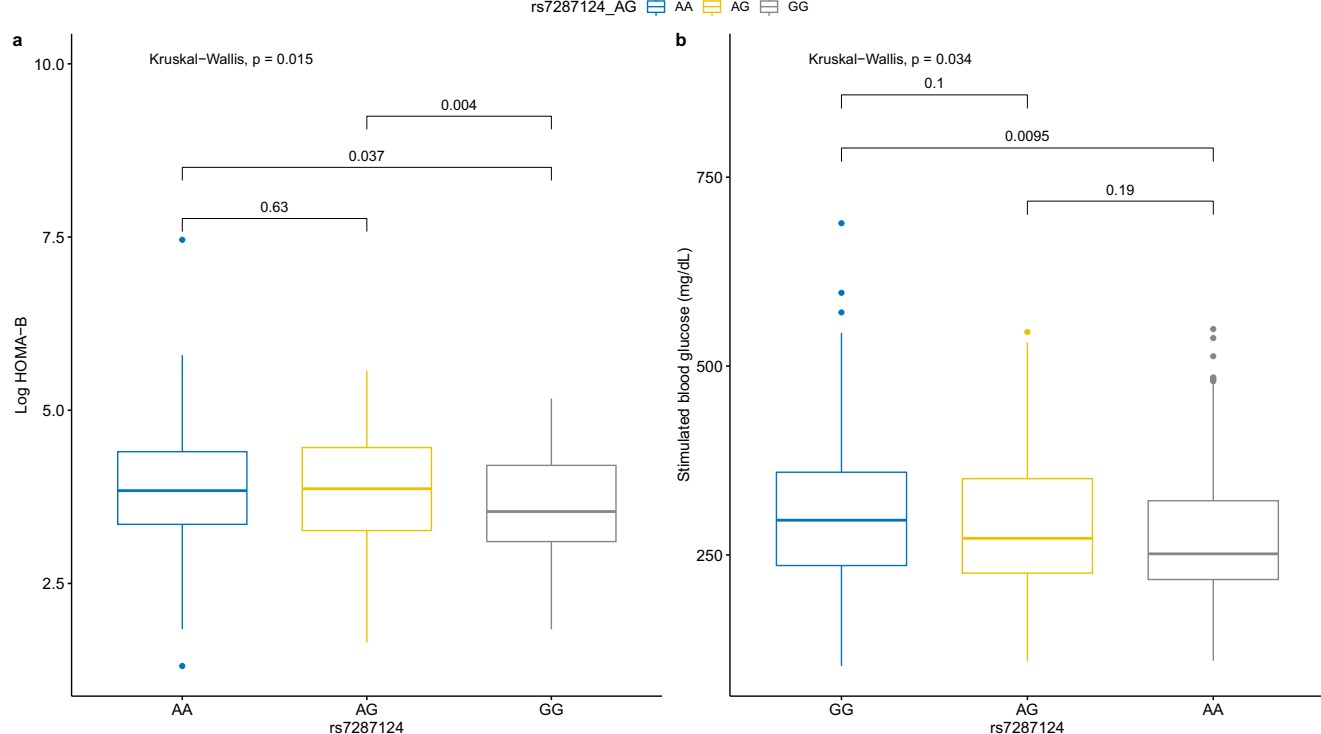

**Fig. 2 | Effect of variant on beta-cell function and 2 hour stimulated glucose levels. a** Box plot showing additive models of XBP1 eQTL variant (rs7287124) and log-transformed Homeostasis Model of Assessment of Beta-Cell function (HOMA-B) calculated using fasted glucose and stimulated C-peptide in 470 individuals with newly diagnosed type 2 diabetes at DMDSC, Chennai, India. After adjustment for age, sex, BMI, and HOMA-B, the association remained significant ($P = 5 \times 10^{-3}$). **b** Boxplot showing effect of rs7287124 on 2 hr stimulated blood sugar in 484 rural-dwelling Indians from the TREND study. After adjustment for age, sex, and BMI, the association remained significant ($P = 0.02$). Per allele was associated with 0.7 mmol/L higher 2 h glucose (95% CI: 0.1–1.4).

no difference in response to GLP-1RA compared to those with the C allele ($P$ value = 0.42), Supplementary Table 6). For sulphonylureas, we found that the allele was associated with a worse response, with each G allele associated with 0.061% (0.7 mmol/mol) lesser reduction in HbA1c (SE: 0.023, $P$ value = 0.008). The variant is independent of all known sulphonylurea pharmacogenetic, dynamic, or kinetic variants (Supplementary Table 7).

## Discussion

Using genetic information, we found that T2DM risk was associated with decreased expression of *XBP1*. An eQTL for *XBP1* reported in pancreatic islets and beta-cells (rs728712(A > G)) was also associated with lower beta-cell function using HOMA-B and higher stimulated glucose in Asian Indians with newly-diagnosed T2DM. In a trans-ancestry meta-analysis, we observed that variant carriers had higher HbA1c at diagnosis ($P$ value $8 \times 10^{-7}$), and that this effect was stronger in individuals diagnosed young with non-obese BMI ($P = 2 \times 10^{-4}$). The variant is more common in those of South Asian (46–50%), East Asian (65–68%), and West African (45%) ancestry compared with Europeans (22–25%). Finally, we detected that for variant carriers, treatment with insulin secretagogue: sulphonylurea was associated with higher post-treatment HbA1c. A visual abstract summarizing the findings can be found Supplementary Fig. 6.

To examine pre-treatment glycaemic control in individuals with new-onset diabetes, we applied strict inclusion criteria around time of diagnosis. Our trans-ancestry meta-analysis showed the average per allele effect of rs7287124 on HbA1c was 4.32 mmol/mol ($P = 8 \times 10^{-7}$). This is equivalent to an 8.64 mmol/mol (0.79%) difference when comparing homozygotes (those with AA v. GG genotypes). Individuals diagnosed young and with non-obese BMI, the per allele effect was 6.41 mmol/mol, which is equivalent to a difference of 12.82 mmol/mol (1.2%) in HbA1c comparing homozygotes.

All else being equal (age at diagnosis, sex, BMI, and ancestry), the per allele difference for rs7287124 was equivalent to or greater than newer

diabetes therapies. A meta-analysis of the efficacy of recently introduced diabetes therapies showed the average reduction in HbA1c in response to DPP4 inhibitors, SGLT2 is, and GLP1-RA was 0.53%, 0.79%, and 0.78% respectively. These differences are more modest than the observed effect of rs7287124 on HbA1c in this study, particularly in the young and non-obese sub-group.

Sulphonylureas, which close the $K_{ATP}$ channel and force beta-cells to secrete insulin independently of glucose concentrations, may place an additional load on the ER and secretory pathway. Moreover, sulphonylureas have been demonstrated to directly induce apoptosis in human islets[45]. We find that reduced islet *XBP1* levels in carriers of rs7287124 likely render beta-cells more sensitive to ER stress and therefore have diminished glycaemic control in response to sulphonylurea therapy. Future studies will be required to investigate the impact of sulphonylurea treatment on beta-cell function, differentiation, and survival in risk allele carriers.

GLP-1RA response is possibly limited by the fact that very few users are on monotherapy. Beyond that, there is evidence that GLP-1RA have an endoplasmic stress-reducing effect[46,47]. This may negate or overcome the diminished ER-stress response associated with reduced *XBP1* expression.

The use of BMI as a surrogate for body composition and high metabolic stress is a limitation. BMI has been shown to be less specific than measures such as waist-to-hip ratio, particularly in South Asians, who are more likely to carry fat ectopically. However, waist-to-hip ratio is not widely measured and, therefore, not available across all study cohorts. Our use of ethnicity-specific BMI thresholds in part mitigates the misclassification of obesity. Furthermore, while we have undertaken robust replication, some well-known resources, such as the UK Biobank, were not used as there are under 2000 individuals of South Asian ancestry and 60 of East Asian.

Another limitation of the study is the use of a candidate gene approach necessitated by a lack of data availability in ancestrally diverse populations. The association observed in our study is similar to those seen in GWA studies; the most recent meta-GWAS of HbA1c demonstrated an effect size

**Fig. 3 | Trans-ancestry meta-analyses of the effect of rs7287124 on HbA1c levels overall and in young, lean T2DM cases.** Forest plots showing random effects of XBP1 eQTL variant on HbA1c (mmol/mol). Effects are pooled across summary statistics available from the MAGIC consortium for East Asians, Europeans, and South Asians without T2DM (n = 173,186) and for 6482 individuals with newly-diagnosed T2DM across 4 cohorts: Tayside Scotland (TDS), urban-dwelling South Asian Indians (DMDSC), rural-dwelling South Asian Indians (TREND), and British South Asian Pakistani and Bangladeshis (G&H). The random effect meta-analysis shows an increase of 4.32 mmol/mol of HbA1c per risk allele (95% CI: 2.60–6.04). Forest plots of sub-groups of people with newly diagnosed T2DM showed a stronger effect in 477 individuals diagnosed young with non-obese BMI (6.41 mmol/mol), compared to no effect in those diagnosed older with non-lean BMI (n = 3382).

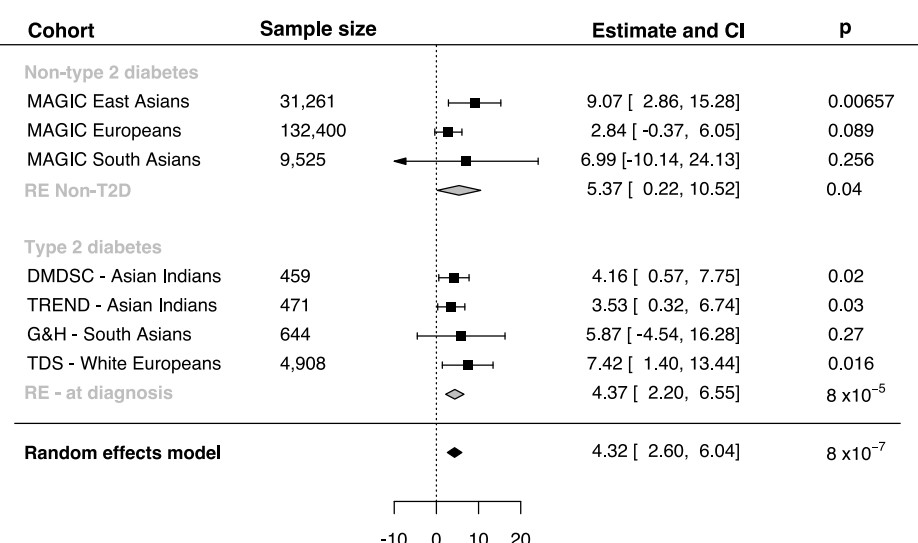

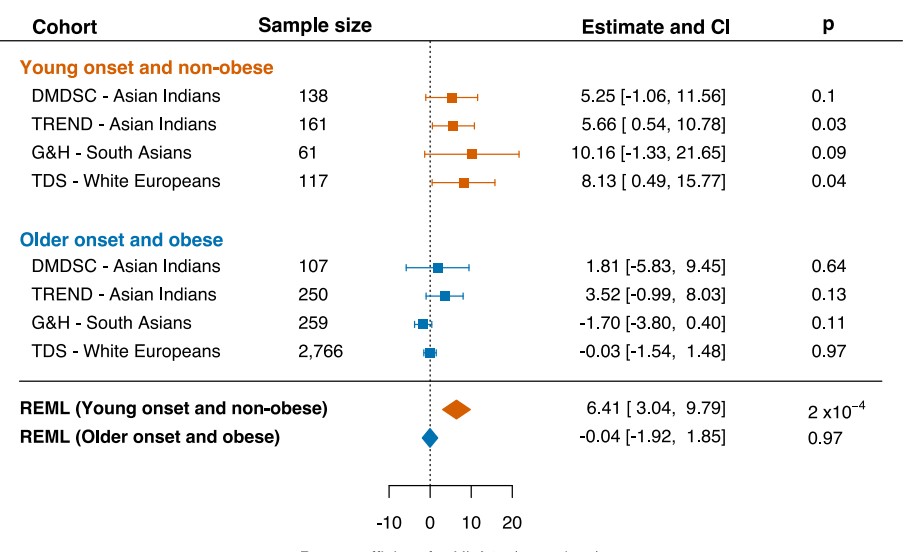

for common variants (such as rs2971669 in *GCK*) with similar allele frequency as rs7287124 (in Europeans), was ~3.3 mmol/mol. However, signals such as *XBP1* are likely to be masked in such GWAS, as research in populations particularly susceptible to insulin deficiency or beta-cell dysfunction (such as South Asians, East Asians, and West Africans) is limited[3][-6,20]. While efforts to increase the representation of non-European ancestries have been growing, studies of glycaemic traits are still more strongly representative of Europeans (70%)[18,20], leaving the remaining 30% to capture genetic effects in all remaining ancestry groups. Similarly, pharmacogenomic studies are almost entirely representation of European cohorts[42,43,48]. Finally, due to the challenge of capturing treatment-naïve glycaemic measures, large-scale meta-GWAS for glycaemic traits are undertaken in healthy individuals who do not have diabetes. The lack of individual-level data for these studies, particularly around the age and BMI at diagnosis, precludes them from subgroup analyses.

Therefore, we utilised a candidate gene approach, building on a strong foundation of in vitro, in vivo, and human single-cell sequencing studies. To support this approach, we validate our target using colocalisation analyses with cell-specific transcriptomics, validating the signal in all available resources of pancreatic islet expression data in humans[23,44]. We observe the same pattern whether using eQTLs in the pancreas or islet cells. The

stronger association with pancreas samples is likely due to the larger single study in GTEx relative to islet-specific transcriptomics.

The lower risk allele frequency in Europeans could be a result of positive selection. Indeed, calculating a gene-based score for recent selection based on data from the 1000 Genomes Selection Browser[49], we find *XBP1* (and a 20 kb region flanking it which includes *ZNRF3*) is ranked in the top 6.8% of all genes and long non-coding RNAs in terms of evidence of recent selection in the European CEU population relative to the Chinese CHB population (details in Supplementary). Selection conversion suggests that protective alleles for increased *XBP1* expression have been selected for to a greater degree in Europeans, who also have a lower risk of beta-cell dysfunction and therefore lower frequency of young and lean onset diabetes. This would explain why risk of lower *XBP1* expression is colocalised with T2DM risk in East Asians rather than white Europeans.

Since rs7287124 is located in *ZNRF3*, we validate its involvement in T2DM by examining evidence for genetic burden of common variants located in the gene using the AMP-T2DM knowledge portal. Using summary statistics of all available GWAS, the Human Genetic Evidence guidelines (HuGE) tool calculates a Bayes Factor of $45 \times 5^{-8}$, concluding that there is very strong evidence for the involvement of variants located in this gene (eQTL and tagger variants for *XBP1* expression) with T2DM risk in

humans (Supplementary Fig. 7)[50]. Interestingly, a test for the well-known beta-cell gene *TCF7L2* produces a similar level of evidence.

Our findings are generalizable to the populations included in the study and have particular value to those with the phenotype of young-onset diabetes in the absence of obesity. People of South Asian descent are nearly twice as likely to have young-onset T2DM and nearly four times more likely to have young onset with lean BMI compared to white Europeans[3,5]. A unique feature of this study is the use of ancestrally diverse populations, and in particular, the dissection of South Asian ancestries. We observe differences across these two South Asian groups in risk allele frequency: in South Indians (DMDSC + TREND), we observe a higher percentage of homozygous risk allele carriers amongst compared to British Pakistanis and Bangladeshis in G&H. The average age of diagnosis, BMI were lower in the South Indians, while HbA1c was higher (Supplementary Table 3). Reflecting this diversity both in location and genetics, we have used different thresholds to classify non-obese BMI in non-migrant Asian Indians compared to migrant South Asians[39].

Our findings suggest that ethnicities such as South Asians have lower genetically-determined *XBP1* expression, resulting in worse beta-cell response to metabolic stress, leading to T2DM. Our findings support the value of genomics in precision diagnosis and therapeutics. Further studies of reversal of glycaemic deterioration for carriers of *XBP1* variant would be useful and could be undertaken using recall-by-genotype trials. The discovery of this association in type 2 diabetes and glycaemic traits in humans highlights the potential for development of strategies to therapeutically enhance *XBP1* expression in pancreatic beta-cells to reverse beta-cell decline and dysfunction.

## Data availability

**Genes and Health**: Individual-level participant data are available to researchers and industry partners worldwide via application to and review by the Genes and Health Executive (https://www.genesandhealth. org/); applications are reviewed monthly. Approved researchers have access to individual-level data in the Genes & Health Trusted Research Environment (TRE) and can request the data files used in this study from the corresponding author(s). All data exports from the Genes and Health TRE are reviewed to prevent release of identifiable individual-level data. Summary data may be exported for cross-cohort meta-analysis or replication and for publication, subject to review. **DMDSC and TREND**: Individual-level participant data are available to researchers and industry partners worldwide via application to and review by the Madras Diabetes Research Foundation. Approved researchers have access to individual-level data in the MDRF data server and can request the data files used in this study from the corresponding author(s). All data exports from the MDRF data server are reviewed to prevent release of identifiable individual-level data. Summary data may be exported for cross-cohort meta-analysis or replication and for publication, subject to review. For access contact dranjana@drmohans.com. **Tayside Diabetes cohort study:** Individual-level participant data are available to researchers via application to https://www. registerforshare.org/share-application-forms and data requests can be emailed to studies@registerforshare.org. For individual-level study specific data requests please contact moneeza.siddiqui@qmul.ac.uk. The source data for Fig. 1 is in Supplementary Data 1. The source data for Fig. 2 cannot be provided as it represents individual-level data. The source data for Fig. 3 is provided in Supplementary Data 2.

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

## Acknowledgements

All volunteers who have facilitated this research across consortia and biobanks. **Tayside Diabetes Cohort** (GoDARTS) has been funded and supported by the WTCCC (072960/Z/03/Z, 084726/Z/08/Z, 084727/Z/08/Z, 085475/Z/08/Z, 085475/B/08/Z) and as part of the European Union Innovative Medicines Initiative SUMMIT program. GoDARTS has been supported by Tenovus Scotland and Diabetes UK grants. SHARE is an NHS Scotland Research infrastructure initiative and is funded by the Chief Scientist Office of the Scottish government. Additional Funding and initiation of the spare blood retention at NHS Tayside were supported by the Wellcome Trust Biomedical Resource Award (099177/Z/12/Z). Additional genome-wide array data were collected with funding from the National Institute for Health Research (INSPIRED [16/136/102]) with use of U.K. aid from the U.K. government to support global health research. We are grateful to all the participants in this study, the general practitioners, the Scottish School of Primary Care for their help in recruiting the participants, and to the whole team, which includes interviewers, computer and laboratory technicians, clerical workers, research scientists, volunteers, managers, receptionists, and nurses. The study complies with the Declaration of Helsinki. We acknowledge the support of the Health Informatics Centre, University of Dundee, for managing and supplying the anonymized data and NHS Tayside, the original data owner. **TREND** and **DMDSC biobank** has been funded by the National Institute for Health Research (INSPIRED [16/136/102]) with use of U.K. aid from the U.K. government to support global health research. The DMDSC biobank includes the MDRF Biobank supported by the Indian Council of Medical Research for the ICMR funded studies. **Genes & Health** is/has recently been core-funded by Wellcome (WT102627, WT210561), the Medical Research Council (UK) (M009017, MR/X009777/1), Higher Education Funding Council for England Catalyst, Barts Charity (845/1796), Health Data Research UK (for London substantive site), and research delivery support from the NHS National Institute for Health Research Clinical Research Network (North Thames). Genes & Health is/has recently been funded by Alnylam Pharmaceuticals, Genomics PLC, and a Life Sciences Industry Consortium of AstraZeneca PLC, Bristol-Myers Squibb Company, GlaxoSmithKline Research and Development Limited, Maze Therapeutics Inc., Merck Sharp & Dohme LLC, Novo Nordisk A/S, Pfizer Inc., and Takeda Development Centre Americas Inc. We thank Social Action for Health, Centre of The Cell, members of our Community Advisory Group, and staff who have recruited and collected data from volunteers. We thank the NIHR National Biosample Centre (UK Biocentre), the Social Genetic & Developmental Psychiatry Centre (King's College London), Wellcome Sanger Institute, and Broad Institute for sample processing, genotyping, sequencing, and variant annotation. We thank: Barts Health NHS Trust, NHS Clinical Commissioning Groups (City and Hackney, Waltham Forest, Tower Hamlets, Newham, Redbridge, Havering, Barking and Dagenham), East London

NHS Foundation Trust, Bradford Teaching Hospitals NHS Foundation Trust, Public Health England (especially David Wyllie), Discovery Data Service/Endeavour Health Charitable Trust (especially David Stables), NHS Digital - for GDPR-compliant data sharing backed by individual written informed consent. Most of all, we thank all of the volunteers participating in **Genes and Health**. For pharmacogenomic GWAS look-ups, we would like to acknowledge the following individuals and consortia. For **Sulphonylurea response** Dr. Sook Wah Yee, Prof. Kathleen M Giacomini, and MetGen plus investigators. For **GLP1-RA response** IMI-DIRECT Investigators & GSK for the use of summary data from the HARMONY trials. This research was funded by the National Institute for Health Research (NIHR) (INSPIRED 16/136/102) using UK aid from the UK Government to support global health research. The views expressed in this publication are those of the author(s) and not necessarily those of the NIHR or the UK Department of Health and Social Care. MKS was funded through the University of Dundee's Baxter Fellowship scheme. The funders were not involved in study design or manuscript preparation. Authors were not precluded from accessing data in the study, and they accept responsibility to submit for publication. MKS and RM are supported by Barts Charity (MGU0504).

## Author contributions

T.D.: conceptualisation, formal analyses, visualisation, writing original draft, and manuscript review. R.M.A. and A.V.: conceptualisation, supervision, data curation, project administration, resources, funding acquisition, manuscript review. S.S., A.Y.D., M.B., A.T., and E.T.A.: data curation, formal analysis, methodology, visualisation, writing and reviewing manuscript. J.S., V.R., & R.P.: data curation, methodology, project administration, resources, manuscript revision. A.S., D.D., and S.H.: data curation, formal analyses, validation, manuscript review. J.C. and N.S.: methodology, manuscript review. R.M. and S.F.: data curation, methodology, supervision, resources, validation, manuscript review. E.R.P., V.M., and C.N.A.P.: funding acquisition, project administration, methodology, resources, supervision, manuscript review. A.A.B.: formal analysis, funding acquisition, investigation, methodology, supervision, visualisation, original draft, and review. M.K.S.: data curation, formal analysis, funding acquisition, investigation, methodology, project administration, resources, supervision, visualisation, writing, and review of manuscript.

## Competing interests

The authors declare no competing interests.

## Additional information

**Moneeza K. Siddiqui** [ID][1,2] ✉, **Theo Dupuis**[3], **Ranjit Mohan Anjana** [ID][2,4], **Adem Y. Dawed**[3], **Margherita Bigossi** [ID][3], **Sundararajan Srinivasan** [ID][3], **Sam Hodgson** [ID][5], **Ebenezer Tolu Adedire**[3], **Alasdair Taylor**[3], **Jebarani Saravanan**[2], **Ambra Sartori**[6], **David Davtian**[3], **Radha Venkatesan**[2], **Alison McNeilly**[5], **James Cantley**[5], **Rohini Mathur**[1], **Naveed Sattar** [ID][7], **Genes & Health Research Team***, **Sarah Finer** [ID][1], **Ewan R. Pearson** [ID][3], **Rajendra Pradeepa**[2], **Viswanathan Mohan**[2,4], **Colin N. A. Palmer**[3], **Andrew A. Brown**[3] & **Ana Viñuela** [ID][3,8]

[1]Wolfson Institute of Population Health, Queen Mary University of London, London, UK. [2]Madras Diabetes Research Foundation, Chennai, India. [3]Division of Population Health and Genomics, University of Dundee, Ninewells Hospital and Medical School, Dundee DD1 9 SY, UK. [4]Dr Mohan's Diabetes Specialities Centre and Madras Diabetes Research Foundation, Chennai, India. [5]Division of Systems Medicine, University of Dundee, Ninewells Hospital and Medical School, Dundee, UK. [6]Department of Genetic Medicine and Development, Geneva University Hospital and University of Geneva Medical School, Geneva, Switzerland. [7]Institute of Cardiovascular and Medical Sciences, University of Glasgow, Glasgow, UK. [8]Biosciences Institute, Faculty of Medicine, Newcastle University, Newcastle Upon Tyne, UK. *A list of authors and their affiliations appears at the end of the paper. ✉e-mail: moneeza.siddiqui@qmul.ac.uk

## Genes & Health Research Team

**Eamonn Maher**[9], **Shabana Chaudhary**[10], **Joseph Gafton**[10], **Karen A. Hunt**[10], **Shapna Hussain**[10], **Kamrul Islam**[10], **Mohammed Bodrul Mazid**[10], **Elizabeth Owor**[10], **Jessry Russell**[10], **Nishat Safa**[10], **John Solly**[10], **Marie Spreckley**[10], **David A. Van Heel**[10], **Jan Whalley**[10], **Ishevanhu Zengeya**[10], **Emily Mantle**[10], **Shaheen Akhtar**[11], **Samina Ashraf**[11], **Dan Mason**[11], **John Wright**[11], **Daniel MacArthur**[12], **Michael Simpson**[13], **Richard C. Trembath**[13], **Gerome Breen**[13], **Raymond Chung**[13], **Sang Hyuck Lee**[13], **Omar Asgar**[14], **Joanne Harvey**[14], **Karen Tricker**[14], **Caroline Winckley**[14], **Hanifa Khatun**[14], **Amna Asif**[14], **Claudia Langenberg**[15],

Grainne Colligan[16], Ceri Durham[16], Bill Newman[17], Ahsan Khan[18], Hilary Martin[19], Teng Heng[9], Matt Hurles[9], Vivek Iyer[9], Georgios Kalantzis[9], Vladimir Ovchinnikov[9], Iaroslav Popov[9], Klaudia Walter[20], Panos Deloukas[21], David Collier[21], Ana Angel[1], Saeed Bidi[1], Fabiola Eto[1], Sarah Finer[1], Chris Griffiths[1], Sam Hodgson[1], Benjamin M. Jacobs[1], Rohini Mathur[1], Caroline Morton[1], Asma Qureshi[1], Stuart Rison[1], Annum Salman[1], Miriam Samuel[1], Moneeza K. Siddiqui[1], Daniel Stow[1], Sabina Yasmin[1], Julia Zöllner[1] & Sheik Dowlut[1]

[9]Aston University, Birmingham, UK. [10]Blizard Institute, Queen Mary University of London, London, UK. [11]Bradford Teaching Hospitals NHS Foundation Trust, London, UK. [12]Garvan Institute, Sydney, Australia. [13]King's College London, London, UK. [14]Manchester University Hospitals, Manchester, UK. [15]Precision Healthcare University Research Institute, Queen Mary University of London, London, UK. [16]Social Action for Health (charity), London, UK. [17]University of Manchester, London, UK. [18]Waltham Forest Council, London, UK. [19]Wellcome Sanger Institute, Cambridge, UK. [20]Wellcome Sanger Institute, London, UK. [21]William Harvey Research Institute, Queen Mary University of London, London, UK.

