## [Transparent Peer Review file · Communications Medicine]

XBP1 expression in pancreatic islet cells is associated with poor glycaemic control especially in young non-obese onset diabetes across ancestries

Corresponding Author: Dr Moneeza Siddiqui

Version 0:

Reviewer comments:

Reviewer #1

(Remarks to the Author)

In the paper “XBP1 expression in pancreatic islet cells is associated with poor glycemic control especially in young non-obese onset diabetes across ancestries,” Siddiqui et. al conduct multiple database analyses to support their hypothesis that XBP1 expression is associated with increased T2DM risk, impaired beta cell function and glycemic control, particularly among ethnic groups that develop this disease at a lower BMI.

While I find this paper interesting, I believe many of the assertions require further experimental proof.

1. It is imperative that the reader understand the strength of evidence to support looking solely at XBP1. However, the key references in this section of the paper are mixed up, while other key statements remain unreferenced.

For example, it is stated that islets of 8 individuals with T2DM showed decreased spliced XBP1 expression, but the referenced paper is clearly wrong. Another key statement mentions “they identified XBP1 as a key transcription factor that was down regulated in individuals with diabetes compared to those with pre-diabetes and healthy controls” but no reference is provided.

2. While rs7287124 is an eQTL for XBP1 in pancreatic islets, it is a robust eQTL for other genes in different tissues. For example, rs7287124 is an eQTL for ZNRF3 and, based on the T2DKnowledge portal, there is very strong evidence that ZNRF3 affects both T2DM and various measures of central obesity which may confound the premise of this study that all associations with this SNP are due to its effect XBP1.

The statement in the results (line 249) “the G allele is associated with lower XBP1 expression and is not an eQTL for any other genes” needs to be modified to “is not an eQTL for any other pancreatic genes.”

3. Line 245- Since rs7287124 and rs58004020 are in very strong LD, clearly more than the single intronic variant rs7287124 can act as a cis eQTL for XBP1 expression in pancreatic islets.

4. Line 252 – there is no Table 2- it appears to have been merged with Table 1

5. Line 265 – “Both variants (rs7287124 and rs58004020) are in the gene ZNFR3 but have no expression on ZNRF3” need to have the modifier “in the pancreas” since they are robust eQTLs in other tissues.

6. Many of the analyses had small sample sizes. Did you perform power calculations?

Reviewer #2

(Remarks to the Author)

This study investigates the role of XBP1, a transcription factor linked to endoplasmic reticulum stress, in the pathogenesis of type 2 diabetes mellitus (T2DM), with a focus on glycemic control and beta-cell dysfunction across diverse ancestries. The authors integrated colocalization analysis, and pharmacogenetics understand the role of XBP1 in glycemic control. The paper is well-structured, leveraging public datasets to explore the interplay between the eQTL rs7287124, XBP1 expression, and diabetes-related phenotypes.

The study identifies rs7287124 as a regulatory variant influencing XBP1 expression in pancreatic islets, with elevated XBP1 levels correlating with poor glycemic control, particularly in young, non-obese individuals with diabetes. Colocalization analysis supports a shared causal variant for both XBP1 expression and T2DM risk, strengthening the genetic link.

Association study reveal that the risk allele at rs7287124 is associated with impaired beta-cell function, as measured by HOMA-B, suggesting a direct role in beta-cell failure. Pharmacogenetic analyses further demonstrate that carriers of the risk allele exhibit diminished therapeutic responses to sulphonylureas, highlighting potential clinical implications.

The study's multi-ancestry design enhances the generalizability of findings, addressing a critical gap in genetic studies often limited to European populations. The integration of colocalization analysis provides compelling evidence for a causal relationship between XBP1 expression and T2DM risk, while the inclusion of pharmacogenetic data adds translational relevance. The focus on young, non-obese individuals with diabetes is innovative, as this subgroup is understudied yet represents a clinically distinct phenotype.

While the study comprehensively links rs7287124 to beta-cell dysfunction, I am just curious whether the components of the HOMA score (fasting glucose and insulin) were individually tested for association with the variant. Additionally, the pharmacogenetic analysis of sulphonylurea response would benefit from controlling for confounding alleles or variants in genes known to influence drug metabolism This would strengthen the claim that rs7287124 itself modulates therapeutic efficacy. Also there is a typo in line 333 for the rsID, I assume it should be rs7287124.

Reviewer #4

(Remarks to the Author)

The genetic factors and pathomechanisms that predispose healthy individuals to metabolic disease including diabetes are not well understood. Here, the authors Siddiqui et al, provide compelling evidence that the reduced of XBP1 expression is associated with poor glycemic control especially in young non-obese onset diabetes. The manuscript is well written and the study is well designed.

XBP1 is a highly conserved transcription factor which control homeostatic and disease-specific UPR. Additionally XBP1 is crucial for proteostasis effects mediated by insulin and other physiological factors like activated protein C, VEGF etc. Therefore, reduced expression of XBP1 affect proteostasis especially in secretory cell types like islets.

In addition to XBP1, mammals cell types express ATF6 and ATF4 which orchestrate UPR together with XBP1. The loss of XBP1 especially should influence the expression of either ATF6 or ATF4. Therefore, showing the expression of ATF6 and ATF4 would provide more insights into the pathomechanims which may affect islets function. A role for ATF6 has been shown to play a role in regulation of islet cell function.

Additionally, from the islet cell gene expression database, additional information related to the genes downstream of XBP1 (sXBP1 target genes) would strengthen this hypothesis.

Minor Comments:

The manuscripts are reviewed by bioinformaticians, clinicians and biologists etc, and therefore, it is helpful when the authors describe all the abbreviations especially technical terms at first place. Also in the figure legends.

On page12 and page 13, the authors describe Figure 4a and 4b. I see only three main figures and I understand that the authors are referring to two parts of Figure.3. Appropriate labeling of figures and figure legends should be done.

Version 1:

Reviewer comments:

Reviewer #2

(Remarks to the Author)

The authors have comprehensively and properly responded to the questions and concerns of the reviewers. I have no further questions.

Reviewer #4

(Remarks to the Author)

I have no further comments

Response to reviewers: XBP1

** Please ensure you delete the link to your author homepage in this email if you wish to forward it to your coauthors **

Dear Dr Siddiqui,

My sincere apologies for the delay in getting back to you with a decision, which was due to challenges in securing the necessary Reviewers, and periods of short staffing. Your manuscript entitled "XBP1 expression in pancreatic islet cells is associated with poor glycaemic control especially in young non-obese onset diabetes across ancestries" has now been seen by 3 referees. You will see from their comments below that while they find your work of considerable interest, some important points are raised. We are interested in the possibility of publishing your study in *Communications Medicine*, but would like to consider your response to these concerns in the form of a revised manuscript before we make a final decision on publication.

We therefore invite you to revise and resubmit your manuscript, taking into account the points raised and addressing them in full. Please highlight all changes in the manuscript text file.

We are committed to providing a fair and constructive peer-review process. Do not hesitate to contact us if you wish to discuss the revision in more detail or if there are specific requests from the reviewers that you believe are technically impossible or unlikely to yield a meaningful outcome.

At the same time, we ask that you ensure your manuscript complies with our editorial policies. Please see our revision file checklist for guidance on formatting the manuscript and complying with our policies. You will also find guidelines for replying to the referees' comments.

Communications Medicine seeks to improve the standards and transparency of reporting in our papers, and to ensure that all submissions conform with the editorial policies of Nature Research. When uploading your revised files please complete and submit Reporting Summary and Editorial Policy checklists as 'checklist' file types. Please note that these forms are a dynamic 'smart pdf' and must therefore be downloaded and completed in Adobe Reader, instead of being opened in a web browser. All points on the checklists must be addressed; if needed, please revise your manuscript in response to these points.

Your revised paper will not be returned to the editors for evaluation until these forms are provided.

Please use the following link to submit your revised manuscript, point-by-point response to the referees' comments (which should be in a separate document to the cover letter), reporting summary, editorial policy checklist and any additional files:

<https://mts-commsmed.nature.com/cgi-bin/main.plex?el=A2DK5BEG3A2tBu3I4A9ftdaV2brmoX4bIUjUXrvbxBKwZ>

We hope to receive your revised manuscript within three months. Please get in touch if you think you might need more time.

Please do not hesitate to contact me if you have any questions or would like to discuss these revisions further. We look forward to seeing the revised manuscript and thank you for the opportunity to review your work.

Best regards,

Andreia Cunha, PhD
Chief Editor
Communications Medicine
Nature Portfolio

Referee expertise:

Referee #1: Epidemiology, T2D diabetes genetics, diverse populations

Referee #2: Genetics, bioinformatics, monogenic diabetes

Referee #3: XBP1, diabetes, molecular and cellular biology

Reviewers' comments:

Reviewer #1 (Remarks to the Author):

In the paper “XBP1 expression in pancreatic islet cells is associated with poor glycemic control especially in young non-obese onset diabetes across ancestries,” Siddiqui et. al conduct multiple database analyses to support their hypothesis that XBP1 expression is associated with increased T2DM risk, impaired beta cell function and glycemic control, particularly among ethnic groups that develop this disease at a lower BMI.

While I find this paper interesting, I believe many of the assertions require further experimental proof.

1. It is imperative that the reader understand the strength of evidence to support looking solely at XBP1. However, the key references in this section of the paper are mixed up, while other key statements remain unreferenced.

For example, it is stated that islets of 8 individuals with T2DM showed decreased spliced XBP1 expression, but the referenced paper is clearly wrong. Another key statement mentions “they identified XBP1 as a key transcription factor that was down regulated in individuals with diabetes compared to those with pre-diabetes and healthy controls” but no reference is provided.

Response: Thank you for noticing this error - we have corrected the reference in text, the citation should be Engin et al.

2. While rs7287124 is an eQTL for XBP1 in pancreatic islets, it is a robust eQTL for other genes in different tissues. For example, rs7287124 is an eQTL for ZNRF3 and, based on the T2DKnowledge portal, there is very strong evidence that ZNRF3 affects both T2DM and various measures of central obesity which may *confound* the premise of this study that all associations with this SNP are due to its effect XBP1.

Response: Thank you for this comment and the reviewer is correct, rs7287124 is an eQTL for ZNRF3 in other tissues. The reviewer is also correct that on a gene level there is strong evidence that common variants in ZNRF3 are associated with obesity traits such as Waist-to-hip ratio adjusted for BMI.

For the first comment, that rs7287124 is an eQTL for other genes (including ZNRF3) in other tissues, we believe this may be due to a misunderstanding of our arguments supporting XBP1 as a candidate gene mediating the activity of rs7287124. XBP1 has been widely reported to mediate beta-cell dysfunction with a consequent role in T2D development. Such an effect has been reported in relation to high fat /metabolically stressful diets, but the main molecular mechanism driven by XBP1 expression changes occur in pancreatic beta-cells and spliced XBP1 has been predominantly found in beta-cells. Therefore, we believe it is reasonable to

focus a search of genetic factors regulating the expression of *XPB1* in beta-cells or as a proxy, on pancreatic islets. Our search led to identification of rs7287124 after performing fine mapping on all possible SNPs associated with **XPB1 expression in islets**. Further analysis supported the hypothesis that rs7287124 is involved in beta cell function, glycaemic control, T2D and its activity is mediated by *XPB1* expression.

We do not claim that rs7287124 is only associated with the expression of *XPB1*, as it is associated with other genes in other tissues. For example, the GTEx catalogue (www.gtexportal.org) reports significant associations of the SNP with *ZNRF3* in artery tissue (pvalue=5.7e-17), and to *XPB1* in whole pancreas (pvalue=2.3e-11) and lung (pvalue=5.5e-8). This is common, as It is well known that SNPs associated with gene expression (eQTLs) can be highly pleiotropic (Aguet et al, Science 2022; Brown et al, 2023) affecting the expression of multiple genes nearby, often, but not always, shared across tissues. However, it is expected that the role of GWAS SNPs on diseases are mediated by genes acting in particular tissues. We believe this was evidence enough to merit further the evaluation performed in this manuscript.

As to the second comment regarding gene-level testing for *ZNRF3*, it is not apparent which of the common variants in the gene are associated with adiposity traits. More specifically, the variant under consideration, rs7287124 was not itself associated with BMI, waist or bioimpedance measures in our data sets or on the knowledge portal. We therefore hope the reviewer would agree that this specific analysis of this variant is not subject to confounding and that if that were the case, our adjustment of analyses for BMI should account for this effect.

The statement in the results (line 249) “the G allele is associated with lower *XPB1* expression and is not an eQTL for any other genes” needs to be modified to “is not an eQTL for any other pancreatic genes.”

Response: Thank you. The text has now been modify to:

*“Using TIGER summary statistics, we confirm that the G allele is associated with lower *XPB1* expression (Table1 & Supplementary Table2) and is not an eQTL for any other genes in the pancreas.”*

3. Line 245- Since rs7287124 and rs58004020 are in very strong LD, clearly more than the single intronic variant rs7287124 can act as a cis eQTL for *XPB1* expression in pancreatic islets.

Response: We thank the reviewer for pointing out that we have inadequately explained the choice of using rs7287124.

rs58004020 was not uniformly imputed in Hg37 for South Asians as seen in the link https://gnomad.broadinstitute.org/variant/22-29257050-A-T?dataset=gnomad_r2_1 where it is classified as “missing”. Furthermore, due to imputation availability of this SNP and LD-pruning of publicly available summary statistics from MAGIC (2021) and DIAMANTE (2022) the variant was also not uniformly available across all ancestry groups. Therefore, we chose rs7287124 as

a proxy variant that was available for all ancestries. We have now added this explanation to the text.

“In East Asians, the variant identified in the colocalisation analyses was rs58004020, however this variant is not present in build 37 of the human genome for south Asians, therefore we identified a proxy variant that is in strong LD with this lead variant and is uniformly available across all ancestry groups and has survived LD-pruning in GWAS summary statistics. The eQTL variant rs7287124 is in LD with rs58004020 in East Asians, Europeans, and south Asians: $R^2=0.99, 0.98, \text{ and } 0.98$ respectively and $D'=1, 0.72, \text{ and } 0.85$ respectively.”

4. Line 252 – there is no Table 2- it appears to have been merged with Table 1

Response: Thank you for pointing this out - this was a typo. It is Table 1.

5. Line 265 – “Both variants (rs7287124 and rs58004020) are in the gene ZNFR3 but have no expression on ZNRF3” need to have the modifier “in the pancreas” since they are robust eQTLs in other tissues.

Response: We have amended the text as requested to make it clear the eQTLs involved XBP1 in pancreatic islets and pancreas.

6. Many of the analyses had small sample sizes. Did you perform power calculations?

Response: The reviewer is likely referring to the sub-group analyses of niche and hard-to-obtain glycaemic measures such as HOMA-B and HOMA-S from non-European populations, presented in Supplementary Table 5. All other analyses have been undertaken the the largest available resources for pancreatic islet eQTLs, T2D meta-GWAS, and trans-ancestry HbA1c meta-GWAS data, and south Asian bioresources.

As the reviewer will know, these granular measures of beta-cell function are not routinely collected and expensive to measure especially in low-resource settings. For example, and context, even the UK Biobank does not measure HOMA-B, insulin levels, C-peptides etc. at scale.

We therefore used all the data available, and it is worth noting that while the numbers are higher in the “older and obese” sub-group it is the smaller “young and non-obese” sub-group that shows the statistically significant result. From a statistical point of view, p values are only “significant” when there is adequate power to detect effects. Therefore, did not perform *post hoc* power analyses but can say that we did have adequate power to detect effects in our smallest sub-groups. A recent article highlights the pitfalls of post hoc power calculations:

<https://onlinelibrary.wiley.com/doi/10.1002/gepi.22464>

Reviewer #2 (Remarks to the Author):

This study investigates the role of XBP1, a transcription factor linked to endoplasmic reticulum stress, in the pathogenesis of type 2 diabetes mellitus (T2DM), with a focus on glycemic control and beta-cell dysfunction across diverse ancestries. The authors integrated colocalization analysis, and pharmacogenetics understand the role of XBP1 in glycemic control. The paper is well-structured, leveraging public datasets to explore the interplay between the eQTL rs7287124, XBP1 expression, and diabetes-related phenotypes.

The study identifies rs7287124 as a regulatory variant influencing XBP1 expression in pancreatic islets, with elevated XBP1 levels correlating with poor glycemic control, particularly in young, non-obese individuals with diabetes. Colocalization analysis supports a shared causal variant for both XBP1 expression and T2DM risk, strengthening the genetic link. Association study reveal that the risk allele at rs7287124 is associated with impaired beta-cell function, as measured by HOMA-B, suggesting a direct role in beta-cell failure. Pharmacogenetic analyses further demonstrate that carriers of the risk allele exhibit diminished therapeutic responses to sulphonylureas, highlighting potential clinical implications.

The study's multi-ancestry design enhances the generalizability of findings, addressing a critical gap in genetic studies often limited to European populations. The integration of colocalization analysis provides compelling evidence for a causal relationship between XBP1 expression and T2DM risk, while the inclusion of pharmacogenetic data adds translational relevance. The focus on young, non-obese individuals with diabetes is innovative, as this subgroup is understudied yet represents a clinically distinct phenotype.

While the study comprehensively links rs7287124 to beta-cell dysfunction, I am just curious whether the components of the HOMA score (fasting glucose and insulin) were individually tested for association with the variant.

Response: We thank the reviewer for so thoroughly understanding the multi-specialty nature of this research. We are especially heartened to see the appreciation for using globally diverse multi-ancestry populations and focusing on young, non-obese type 2 diabetes.

In response to the question, regarding association of the variant with the components of the HOMA score - specifically fasting glucose was modestly associated with the variant. We however chose to present results from HOMA-B which is a measure of insulin secretion which is the proposed pathway of *XBP1* impact, and this way were able to adjust models of insulin sensitivity (HOMA-S). Given the small numbers in this analysis and the modest p values, we were concerned about being asked to correct for multiple testing if we produced too many findings.

Additionally, the pharmacogenetic analysis of sulphonylurea response would benefit from controlling for confounding alleles or variants in genes known to influence drug metabolism. This would strengthen the claim that rs7287124 itself modulates therapeutic efficacy. Also there is a typo in line 333 for the rsID, I assume it should be rs7287124.

Response: We thank the reviewer for this excellent suggestion that allowed us to assess if any pharmacogenetic or pharmacokinetic (PG and PK) variants associated with sulphonylurea response are in LD with our variant (rs7287124) and therefore confounding the association. As you can see, none of the pharmacogenetic variants are in linkage with rs7287124 (in EUR populations who were part of these pharmacogenetic studies).

The MEtGEn consortium that produced the first GWAS of sulphonylurea response (Dawed et al. <https://doi.org/10.2337/dc21-1152>) consisted of 8 contributing studies the provided summary statistics of the response phenotype which were then meta-analysed. Therefore, in order to re-adjust any of these pharmacogenetic findings for other covariates (clinical or genotype) it would require a new project license be established with MetGen so that all centres can re-run the GWAS with the new suggested models. This would require years of time, but luckily is not necessary given the table below.

We have added text in the manuscript to say:

The variant is independent of all known sulphonylurea pharmacogenetic, -dynamic or -kinetic variants (Supplementary Table 7).

SU PK/PD variant	rs7287124
CYP2C9*2 (Arg144Cys) (rs1799853)	R2=0.0016 D'=0.051
CYP2C9*3 (Ile359Leu) (rs1057910)	R2=0.0 D'=0.001
CYP2C8*3 (linked polymorphisms of Arg139Lys and Lys399Arg)(rs10509681)	R2=0.0015 D'=0.0543

SLCO1B1 521T>C(Val174Ala) (rs4149056)	R2=0.0004 D'=0.0897
CYP2C19*2 (681 G>A) (rs4244285)	R2=0.0011 D'=0.1538
CYP2C19*3 (636 G>A) (rs4986893)	R2= N.A. D'=N.A Linkage equilibrium
KCNJ11 (E23K, rs5219)	R2=0.0 D'=0.0158
ABCC8 (S1369A, rs757110)	R2=0.0 D'=0.0027
TCF7L2 (rs12255372)	R2=0.0045 D'=0.0769

Reviewer #4 (Remarks to the Author):

The genetic factors and pathomechanisms that predispose healthy individuals to metabolic disease including diabetes are not well understood. Here, the authors Siddiqui et al, provide compelling evidence that the reduced of XBP1 expression is associated with poor glycemic control especially in young non-obese onset diabetes. The manuscript is well written and the study is well designed.

XBP1 is a highly conserved transcription factor which control homeostatic and disease-specific UPR. Additionally XBP1 is crucial for proteostasis effects mediated by insulin and other physiological factors like activated protein C, VEGF etc. Therefore, reduced expression of XBP1 affect proteostasis especially in secretory cell types like islets.

In addition to XBP1, mammals cell types express ATF6 and ATF4 which orchestrate UPR together with XBP1. The loss of XBP1 especially should influence the expression of either ATF6 or ATF4. Therefore, showing the expression of ATF6 and ATF4 would provide more insights into the pathomechanisms which may affect islets function. A role for ATF6 has been shown to play a role in regulation of islet cell function.

Additionally, from the islet cell gene expression database, additional information related to the genes downstream of XBP1 (sXBP1 target genes) would strengthen this hypothesis.

Response: The suggestion by the reviewer implies evaluating the effect of rs7287124 on the expression of genes downstream *XBP1*, such as *ATF4* and *ATF6*. The underlying hypothesis being that gene expression differences in *XBP1* driven by genotype differences would have an equivalent effect on genes downstream. These can be identified performing trans-eQTL analyses, where genes are tested for associations with SNPs all over the genome (*trans-*), not just in the vicinity of the gene (*cis-*). Such analyses are very computationally intensive, performing ~9million test per gene for the ~20k genes, and require sample sizes in the order of thousands of samples to identify significant associations, as reported by Vösa et al (Nat. Gen. 2018). Such datasets are not currently available for pancreas or pancreatic islets. To illustrate our point, we used the GTEx catalogue again to evaluate possible associations between rs7287124 and the expression of ATF4 and ATF6, with both tests not identifying significant associations in any tissue, pvalues 0.85 and 0.34 respectively.

However, this comment is synergistic with the author's ideas about possible next steps and we have included a statement in the manuscript to highlight this possible future experiments and resources.

“”

Minor Comments:

The manuscripts are reviewed by bioinformaticians, clinicians and biologists etc, and therefore, it is helpful when the authors describe all the abbreviations especially technical terms at first place. Also in the figure legends.

Response: We thank the reviewer for this comment, indeed this is a multi-specialty paper. We have now combed through the next to ensure all technical terms have descriptions at first instance. These are highlighted in yellow. We have included and signposted a table with such abbreviations, full forms and a brief description (Supplementary Table 1). We've specifically called out this table at the start of the methods section so readers can use refer to it more successfully.

On page12 and page 13, the authors describe Figure 4a and 4b. I see only three main figures and I understand that the authors are referring to two parts of Figure.3. Appropriate labeling of figures and figure legends should be done.

Response: This has been corrected now. Thank you for pointing this out.